# Deciphering the Cardiovascular Potential of Human CD34^+^ Stem Cells

**DOI:** 10.3390/ijms24119551

**Published:** 2023-05-31

**Authors:** Anne Aries, Céline Zanetti, Philippe Hénon, Bernard Drénou, Rachid Lahlil

**Affiliations:** 1Institut de Recherche en Hématologie et Transplantation (IRHT), Hôpital du Hasenrain, 87 Avenue d’Altkirch, 68100 Mulhouse, France; 2CellProthera, 12, Rue du Parc, 68100 Mulhouse, France; phenon@cellprothera.com; 3Groupe Hospitalier de la Région de Mulhouse Sud-Alsace, Hôpital E. Muller, 20 Avenue de Dr Laennec, 68100 Mulhouse, France

**Keywords:** CD34^+^ cells, umbilical cord blood, gene expression profiling, cardiovascular differentiation, cell therapy

## Abstract

Ex vivo monitored human CD34^+^ stem cells (SCs) injected into myocardium scar tissue have shown real benefits for the recovery of patients with myocardial infarctions. They have been used previously in clinical trials with hopeful results and are expected to be promising for cardiac regenerative medicine following severe acute myocardial infarctions. However, some debates on their potential efficacy in cardiac regenerative therapies remain to be clarified. To elucidate the levels of CD34^+^ SC implication and contribution in cardiac regeneration, better identification of the main regulators, pathways, and genes involved in their potential cardiovascular differentiation and paracrine secretion needs to be determined. We first developed a protocol thought to commit human CD34^+^ SCs purified from cord blood toward an early cardiovascular lineage. Then, by using a microarray-based approach, we followed their gene expression during differentiation. We compared the transcriptome of undifferentiated CD34^+^ cells to those induced at two stages of differentiation (i.e., day three and day fourteen), with human cardiomyocyte progenitor cells (CMPCs), as well as cardiomyocytes as controls. Interestingly, in the treated cells, we observed an increase in the expressions of the main regulators usually present in cardiovascular cells. We identified cell surface markers of the cardiac mesoderm, such as kinase insert domain receptor (KDR) and the cardiogenic surface receptor Frizzled 4 (FZD4), induced in the differentiated cells in comparison to undifferentiated CD34^+^ cells. The Wnt and TGF-β pathways appeared to be involved in this activation. This study underlined the real capacity of effectively stimulated CD34^+^ SCs to express cardiac markers and, once induced, allowed the identification of markers that are known to be involved in vascular and early cardiogenesis, demonstrating their potential priming towards cardiovascular cells. These findings could complement their paracrine positive effects known in cell therapy for heart disease and may help improve the efficacy and safety of using ex vivo expanded CD34^+^ SCs.

## 1. Introduction

Severe heart failure develops most often as a result of myocardial infarction (MI) and is characterized by the massive loss or damage of cardiomyocytes. Human autologous CD34^+^ SCs injected into patients with acute MIs may contribute to the repair of an infarcted myocardium by inducing neo-cardiac vascularization, angiogenesis, and the inhibition of cell death [1,2,3,4]. Indeed, the presence of a subpopulation having characteristics of both immature and mature endothelial and cardiomyocyte progenitors was previously identified among mobilized human peripheral blood CD34^+^ cells in cardiac patients [5]. In a previous pilot study, we showed that an intra-cardiac reinjection of autologous peripheral blood CD34^+^ cells collected by leukapheresis and purified by immuno-selection for patients with acute MIs induced significant functional and structural regeneration of cardiac lesions [5]. In a similar study, an intracoronary administration of higher doses of autologous CD34^+^ cells in patients with left ventricular dysfunction after ST-segment elevation myocardial infarction was associated with improvement in left ventricular function and reduction in infarct size [3]. In addition, the epigenetic reprogramming of human CD34^+^ cells significantly induced their myocardial repair capabilities by improving both neo-cardiomyogenesis and neo-vascularization [6]. These data highlight that the use of CD34^+^ SCs is an attractive strategy to repair damaged myocardial tissue in patients after MIs [7]. Nevertheless, the benefit of CD34^+^ SCs therapy is mainly limited by low cell availability [5,8], underlining the need for successful ex vivo expansion to overcome the limited doses of available SCs [9,10]. Therefore, our group recently developed an automated device allowing ex vivo CD34^+^ SCs expansion at the clinical scale after MIs [11,12]. 

Another debate over the direct contribution of CD34^+^ SCs to cardiac tissue regeneration has arisen regarding the mechanism(s) of action by which they promote therapeutic effects [13,14,15,16]. Nevertheless, while this can occur, there is no convincing evidence yet that CD34^+^ SCs have the potential to become cardiac progenitors and cardiomyocytes, thereby contributing directly to the recovery of the infarct area [17,18]. Numerous reports have proposed that the benefits provided by transplanted SCs in cases of heart disease are also likely related to the paracrine secretory effects of soluble factors and exosomes [19,20,21,22,23]. 

Thus, in order to test the possible direct contribution of human CD34^+^ cells in cardiac repair, the main challenge is to determine whether these SCs have the ability to differentiate along the cardiac pathway. Identifying proteins and surface molecules that are known to be associated with the early stages of cardiovascular differentiation of CD34^+^ SCs could be a demonstration of their cardiovascular reprogramming aptitude and, therefore, help in their isolation and characterization.

In order to define whether CD34^+^ SCs have the potential for cardiovascular differentiation, we perform a comprehensive, whole-genome study based on a transcriptomic approach with human cord blood CD34^+^ cells at sequential stages of in vitro differentiation in the cardiovascular pathway using a combination of treatment with epigenetic-modifying agents followed by transforming growth factor (TGF-β) and vitamin C stimulation. Comparative expression profiling between the treated CD34^+^ SCs and those in the basal state reveals that the expressions of the main regulators usually present in cardiovascular cells are induced in treated cells. Interestingly, we observe the expressions of characteristic markers of the cardiac mesoderm and identify the implicated signaling pathways, attesting that they can be initiated towards cardiovascular differentiation.

## 2. Results

### 2.1. Induction of Human CD34^+^ Stem Cells toward Cardiovascular Lineage

To investigate whether undifferentiated CD34^+^ SCs had the ability to differentiate into cardiovascular lineage and, therefore, provide supporting evidence for their potential therapeutic efficiency for use in cardiac regenerative medicine, we performed their induction to cardiovascular differentiation in parallel with that of CMPCs, as described in the Section 4 and summarized in Figure 1A. The observation of the cells via microscope on day 3 of differentiation (Diff d3) showed distinct morphological changes. The CD34^+^ SCs appeared rounded in suspension and gathered in clumps. Interestingly, after around 10 days of culturing in the sequential presence of valproic acid (VPA), 5′azacytidine (5′AZA), and TGF-β1, the differentiated CD34^+^ SCs became adherent and showed ‘cobblestone’ monolayer cell morphologies and structures with short cellular extensions (Figure 1B). These adherent cells continued to proliferate until day 14 (Diff d14) and appeared thin, elongated, and in clusters.

In addition to the CD34^+^ SCs induced to cardiovascular differentiation, we decided to similarly collect human control cell samples from cardiac progenitors at different cardiac stages of differentiation, further establishing a transcriptional signature reference of cardiac differentiation. For this purpose, CMPCs were isolated from human heart tissue biopsies usually discarded during heart surgery. These cells were subjected to cardiac lineage differentiation, as reported previously by Smits et al. [24] and described in the Section 4 (Figure 1C). Distinct morphological changes were observed while the cells differentiated into cardiac lineage. On day 1, the treated CMPCs presented small, spindle shapes (Figure 1C). During the differentiation process, cells formed circular patterns and grew in multiple layers, as shown in the lower panel on day 29 of differentiation. 

We then looked at the stages of differentiation of the CD34^+^ SCs and their treated counterparts of Diff d3 and Diff d14 cells in comparison to CMPCs and cardiomyocytes by analyzing the expressions of some known specific vascular and cardiac marker genes using real time RT-PCR. As shown in Figure 2, the expressions of the cardiac genes GATA-4, HAND2, and TBX20 became significantly induced after 14 days of treatment (231-, 16,112-, and 4503-fold, respectively ** *p* < 0.01), despite the fact that they do not reach levels of those observed in the cases of CMPCs and cardiomyocytes. In addition, KDR and VE-Cad expressions were significantly increased on day 14 (631- and 100-fold, respectively), and so were the positives controls. The expression of desmin remained moderate in Diff 3 and 14, as in CMPCs, and not significantly different from CD34^+^ SCs, despite being strongly induced in cardiomyocytes (Figure 2).

### 2.2. Differential Gene Expression Profiling Analysis

In order to follow and dissect the molecular profiles of these CD34^+^ SCs undergoing cardiovascular pathways differentiation, we studied their genome-wide expression changes to identify new surface markers, signaling pathways, and novel networks that could attest to their cardiovascular specification. We performed differential transcriptome-wide expression profiling between human undifferentiated CD34^+^ SCs and those induced at two stages of differentiation, i.e., Diff d3 and Diff d14, using microarrays. We included CMPCs and human cardiomyocyte RNA samples as positive controls. Four biological replicates from independent experiments were used. Through this study, after the normalization and filtration of the set of 50,599 biological probes that were contained in the arrays, around 28,008 were retained for the analysis. Among these genes, by applying one-way ANOVA, *p* < 0.001, and using the Benjamini-Hochberg method and post hoc Student-Newman-Keuls (SNK) test, we found that around 14,618 were considered significantly expressed (either up- or downregulated). To characterize their transcriptional differences, we performed a statistical study of differential gene transcription expressions between the untreated CD34^+^ SCs and those induced to differentiate on days 3 and 14, as well as CMPCs and cardiomyocytes. As shown in the table of Figure 3A, the total number of differentially expressed genes increased during the induction of differentiation from 3537 on d3 to 7069 on d14, with 8918 genes in the CMPCs, as well as 8702 in cardiomyocytes, revealing significant transcriptome change throughout the stages of differentiation. It, thus, suggests the appearance of a more specialized gene expression phenotype. However, both the up- and downregulated gene numbers increased during the differentiation stages compared to untreated CD34^+^ SCs (Figure 3A). 

Unsupervised hierarchical clustering by Pearson correlation of the microarray samples further confirmed the high reproducibility of transcriptional signatures within groups (Figure 3B). Moreover, the clustering of each sample also revealed the presence of two transcriptomic sets, representing either the undifferentiated (CD34^+^ SCs, Diff d3) or differentiated stages (Diff d14 and positive controls). Untreated CD34^+^ cells were more transcriptionally related to the earliest stage of differentiation of the induced CD34^+^ cells (Diff d3) than they were to CMPCs or cardiomyocytes (Figure 3B). In contrast, CD34^+^ cells induced at the later stage of differentiation (Diff d14) were closer to CMPCs and cardiomyocytes in terms of expression pattern. In addition, as shown in Figure 3C, the gene expression scatter plots representing the normalized log_2_ fold-change in transcripts for Diff d14 cells were compared to untreated CD34^+^ SCs, which showed more changes in gene expression, i.e., 3396 genes were up-regulated, and 3700 genes were down-regulated (upper panel). In contrast, CMPCs versus Diff d14 cells (lower panel) showed less change, indicating that our process of inducing cardiovascular differentiation in vitro was most likely effective. 

### 2.3. Potential Gene Signature Involved in CD34^+^ Cardiovascular Differentiation

An analysis of gene panel variation according to function and possible implication in cardiovascular cell development showed that some sets of gene expressions were affected in the treated cells induced to cardiovascular differentiation. These genes included cell receptors, transcription factors, and genes that controlled proliferation and the cell cycle. Thus, the pluripotency marker genes, such as DNMT3B, TERT, DPPA4, and epithelial cell adhesion molecule (EPCAM), as well as the stem cells receptors CD34 and promin1 (CD133), remained enriched on Diff d3 in comparison to untreated CD34^+^ SCs controls, but they became downregulated at Diff d14 and for CMPCs and cardiomyocytes. In contrast, the expressions of mesodermal and cardiac mesodermal genes, such as neural cell adhesion molecule NCAM1, MESP2, and cardiac troponin I (TNNI3), were found to be enriched as soon as on day 3, while, other genes, such as KDR, NR2F2, and FZD4, became enriched at Diff d14, as in CMPCs and cardiomyocytes (Figure 4). The expressions of cardiac-specific transcription factors, including HAND2, GATA4, GATA6, TBX5, and TBX20 [25], were also found to be enriched at a relatively significant (*p* < 0.001) level in Diff d14 cells compared to CD34^+^ SCs, as well as in the two groups of CMPCs and cardiomyocytes, indicating the possible establishment of cardiac differentiation on day 14 of treatment. Finally, the fact that angiogenesis-enhancing gene expressions of VWF, PECAM1, EFNB2, ENG, NOS3, and CDH5 also showed significant upregulation at Diff d14 (Figure 4) suggests that CD34^+^ SCs in our conditions of differentiation were able to turn on specific genes implicated in the emergence and rising of cardiovascular phenotypes and were thus primed to differentiate into a cardiovascular lineage.

We further sought to investigate the significance of the observed differential gene expression profile between Diff d14 and CD34^+^ SCs by performing a *t*-test (*p* < 0.001) with Benjamini and Hochberg’s correction (Figure 5). Gene ontology analysis of the upregulated genes in Diff d14 versus CD34^+^ SCs illustrated that, in addition of the endothelial cell differentiation genes, an enrichment of genes related to angiogenesis, cardiovascular, and heart development was observed (Figure 5A). As shown in Figure 5B–D) a significant induction of the expression of most cardiac-specific genes—BMP2 (pAdj = 4.89 × 10^−6^), EDN1 (pAdj = 1.13 × 10^−6^), GATA-6 (pAdj = 7.84 × 10^−7^), TBX3 (pAdj = 2.14 × 10^−8^), PDLIM5 (pAdj = 1.14 × 10^−9^) and endothelial: NRP1 (pAdj = 2.81 × 10^−9^), KDR (pAdj = 2.84 × 10^−5^), NR2F2 (pAdj = 2.06 × 10^−9^), SOX17 (pAdj = 3.44 × 10^−11^), NRG1(pAdj = 4.63 × 10^−5^) and cardiovascular: LAMA4 (pAdj = 2.09 × 10^−8^), SERPINE1 (pAdj = 7.64 × 10^−10^), ANGPTL4 (pAdj = 7.84 × 10^−7^)—was found in Diff d14 when compared to CD34^+^ SCs, confirming their potential priming towards endothelial and cardiovascular lineages.

### 2.4. Potential Signaling Pathways Involved in the Initiation of CD34^+^ Cardiovascular Differentiation

In order to identify early signaling pathways involved during the initiation of CD34^+^ SCs differentiation into a cardiovascular lineage, we performed a pathway analysis on the enriched genes at the 14-day stage of differentiation (Diff d14) versus CD34^+^ SCs using the KEGG pathway database. By selecting some genes that showed significantly induced expression differences of at least 1.5-fold (*p* < 0.001), we identified a list of potential candidate genes, which allowed us to draw possible networks. Thus, interestingly, we found enrichment in transcripts belonging to the signaling pathways of Wnt and TGF-β in the CD34^+^ SCs induced to differentiate (Figure 6). Through the analysis of the Wnt-signaling pathway, we observed that SOX17, FOSL1, DKK1, SFRP1, CCND1, and FZD4, -6, and -8 became induced in the differentiated CD34^+^ SCs to the same extent as CMPCs and cardiomyocytes at day 14 (Figure 6A). Interestingly, an increase in gene expressions involved in the specification of the cardiac mesoderm, such as FZD5, FZD3, TCF7, and WNT1, WNT6, and WNT11 [26,27,28], is instigated on day 3, which suggests that their involvement would be early and prior to the initiation of differentiation (Figure 6A). In comparison, we uncovered that WNT11, known to induce cardiac lineage commitment in unfractionated bone marrow mononuclear cells [29], was particularly highly expressed on day 3, while FZD4, a key regulator in cardiac development [28], was the most strongly expressed on day 14 (Figure 6A). 

Furthermore, we also identified significant enrichment in the expressions of Wnt antagonists, such as SOSTDC1, DKK1, and SFRP1, on days 3 and 14 (Figure 6A), suggesting that they might promote cardiovascular differentiation from CD34^+^ cells, as it was shown that these factors were necessary for the downregulation of Wnt signaling and, subsequently, cardiac specification [30].

On the other hand, TGF-β pathways were shown to be involved during early cardiogenesis and the cardiovascular differentiation process [31]. Our results showed increased expressions of BMP family members (BMP2 and BMP4), SMAD6, and INHBA on day 14, which are known to be capable of leading to cardiac specification (Figure 6B). Interestingly, PITX2 and ID1, two transcription factors fundamental to heart development [32] and essential for early heart formation [33], appeared higher during the Diff d3 and Diff d14 stages of differentiation, respectively (Figure 6B).

### 2.5. Initiation to the Cardiovascular Differentiation Evidence for CD34^+^ SCs

In addition to the phenotypic and intracellular modifications of gene expressions, the cardiovascular differentiation of SCs is accompanied by a set of modifications, such as the appearance or reduction in extracellular receptor expressions. Thus, using flow cytometry (FCM), we followed the expressions of cardiovascular cell surface markers in untreated CD34^+^ SCs and in Diff d14 cells, respectively, in the CD34^+^ fraction and in the CD105^+^ population (endoglin), which is a surface marker mainly expressed on endothelial cells and fetal cardiac myocytes [34]. Interestingly, the endothelial surface markers KDR, CD73, CD146, and CD106 were significantly more expressed in the cells induced to differentiate (Figure 7). The cardiac mesoderm receptor CD344 (FZD4) was moderately induced in Diff d14 cells, as well as CD172 (SIRPA), a cardiac marker, and CD117 (c-KIT), an early cardiac marker, which show sensible expression changes. The untreated cells expressed CD31, and its expression was sustained after differentiation. In contrast, CD133 receptor expression decreased in the differentiated cells. We also analyzed the expression of Thy-1, known as CD90 (a mesenchymal marker), which did not show any changes in its expression. These data support the genetic changes observed in differentiated cells and confirmed cardiovascular modifications on differentiated cells without the acquisition of mesenchymal phenotypes.

We then used immunofluorescence staining to follow the expression of some cardiovascular markers in Diff d14 cells. Immunostaining analysis of CD34^+^ SCs population after 14 days of differentiation revealed positive expression of FZD4, CD31, vWF, and sarcomeric α-actin, which consolidates our belief in their initiation towards early cardiovascular differentiation (Figure 8). 

## 3. Discussion

In the present study, we established whether small, epigenetic-repressive markers (VPA and 5′AZA), followed by cardiogenic agent stimulation (TGF-β and vitamin C), could induce the activation of vascular and cardiogenic genes in untreated CD34^+^ SCs. Our results demonstrated that some genes implicated in cardiovascular processes were able to be primed in treated cells compared to untreated ones in as little as three days, with some others, in fourteen days, achieved almost the same RNA phenotypes as CMPCs and cardiomyocytes, indicating the possible ability of untreated CD34^+^ SCs to initiate to a myocardial lineage. Indeed, 5′AZA alone is known to induce the cardiac differentiation of P19 embryonic SCs [35]. Furthermore, it was reported that VPA and 5′AZA conferred cardiac potential to human CD34^+^ cells in vivo [6]. In contrast, VPA alone had an inhibitory effect on the differentiation of committed endothelial-colony-forming cells into functional endothelial cells [36]. Thus, the success of cardiac regeneration during stem cell therapy relies in part on understanding the processes of cardiovascular differentiation and the identification of markers, which could determine whether CD34^+^ stem cells possess the characteristics necessary for differentiation into cardiomyocytes. This could be based on identifying the molecular and cellular signatures of stages of CD34^+^ SCs differentiation.

A global transcriptional analysis of CD34^+^ SCs, induced at various stages of cardiovascular differentiation, has not yet been documented. Thus, to determine their cardiovascular capacity and to identify genes that could predict the cardiac potential of CD34^+^ cells before their clinical use in regenerative medicine, we focused on key signaling pathways that play critical roles in cardiovascular development. Global gene expression profiling and gene ontology analyses were performed on genes that were initially poorly expressed in CD34^+^ SCs compared to CMPCs, but which increased gradually at both stages of differentiation (Diff d3 and diff d14), similar to increases observed in CMPCs and cardiomyocytes, which provided more insight into the expressions of both early and late key regulator genes, as well as on the molecular signaling involved in CD34^+^ SCs commitment to a cardiovascular lineage. Interestingly, on day three after treatment, the de novo mesoderm early cardiac-specific gene transcripts NCAM1, MESP2, cardiac, and troponin I became upregulated, followed, on day 14, by KDR, FZD4, PDGFRα, and transcription factor NR2F2, which has a critical role in controlling the development of the heart and blood vessels [28,37]. However, the troponin level decreased on Diff d14, suggesting that the in vitro cardiac differentiation conditions had limits or needed to be further improved to maintain the expressions of immature cardiomyocyte gene characteristics. In parallel, on Diff d14, we observed a significantly high induction of early cardiac transcription factor gene [25] expressions, including GATA-4, GATA-6, TBX5, and TBX20, compared with untreated CD34^+^ cells. 

The Wnt pathway genes appeared to be activated in differentiated cells, consistent with the essential role of this pathway in controlling the number of cardiac progenitors [38]. With cultured pluripotent stem cells, activation of the Wnt pathway was sufficient for cardiac mesoderm induction [39]. Our findings that Wnt11 was upregulated via FZD4 are also consistent with the results of a previous study, indicating that they regulate the cardiomyocyte differentiation of human embryonic stem cells via noncanonical Wnt signaling [27]. Additionally, other results showed that the ectopic overexpression of Wnt11 in bone marrow mononuclear cells was sufficient to induce cardiac lineage in a protein-kinase-C-dependent manner [29]. However, its expression dosage and timing are important: cardiac fields expand when Wnt is exogenously activated during early gastrulation, but subsequent cardiac crescent formation is inhibited when Wnt is activated during later stages of gastrulation [40,41].

Using pathway analysis, we also found that the TGF-β-signaling pathway could represent a key step during the early cardiovascular lineage commitment of CD34^+^ SCs. Several transcription factors were identified as upregulated in Diff d14 cells, including ID1 and ID3, which are essential for the specification of mesoderm cells into cardiac progenitors during heart formation, as well as in the process of endothelial differentiation [33,42]. Indeed, ID proteins are among the most highly regulated downstream targets of BMP/Smad signaling [43], and their importance was recently reported, as the loss of ID1 and ID3 expression associated with BMPR2 mutations contributed to cardiomyocyte dysfunction in patients with congenital heart disease [44]. Furthermore, other constituents in TGF-β signaling, including the cell surface marker BMPR2, BMP cytokines (BMP2 and BMP4), and even SMAD proteins, could play a role in the initiation of the cardiovascular ability of CD34^+^ SCs, as they showed upregulation on day 14 when cells became primed into cardiovascular lineage. Finally, another key regulator of cardiac development, PITX2, required for cardiac cell proliferation and ventricular development [45], was also upregulated in Diff d14 cells, supporting its possible implication in CD34^+^ SC cardiac commitment. Taken together, all these data clearly highlight a possible important role for TGF-β and Wnt signaling in the initiation of cardiovascular differentiation of CD34^+^ cells. However, these results need to be confirmed by blocking or using CD34^+^ SCs deficient in these two pathways.

However, although our finding is in agreement with others showing a differentiation of bone marrow cells towards cardiomyocytes [46,47], other mechanisms involving CD34^+^ SCs in cardiac repair could not be excluded, as their production of cytokines, growth factors, exosomes, and microparticles is known to decrease apoptosis, inflammation, and fibrosis and to stimulate angiogenesis, thus contributing to improving cardiac function [16,19,48]. 

Cardiac repair potentially induced by CD34^+^ cells delivered intramyocardially as a therapy for myocardial infarctions could, thus, be the result of three intricate mechanisms. 

Injected CD34^+^ SCs are first activated by a mix of cardio-active chemokines secreted by the inflammatory scar [49,50].

Once activated, they are chemoattracted toward home in the cardiac ischemic zone, where they begin to multiply and commit along the endothelial and cardiac pathways [5,51,52]. The very small embryonic-like stem cells, which constitute a rare and quiescent pluripotent fraction of CD34^+^ SCs, are known to support vessel formation [53] and be specified to cardiomyocytes, neurons, and hematopoietic stem cells [54], which can play a role in these processes likely contributing to the repair of an infarcted myocardium.

At the same time, they likely release soluble paracrine factors and exosomes that can enhance resident cardiomyocyte proliferation [54] or support neo-angiogenesis [21], respectively, thus reducing fibrosis and attenuating the alteration effects of AMIs (Figure 9).

## 4. Materials and Methods

### 4.1. Source of Human CD34^+^ SCs

Hu-UCB samples were obtained from healthy, consenting donors undergoing normal deliveries according to procedures approved by the Ethics Review Board of Besançon Hospital (Comité d’éthique et de déontologie (CED), Etablissement Français du Sang (EFS)). 

### 4.2. Human CD34^+^ Cell Culturing and Differentiation

Hu-UCB mononuclear cells were first collected by Ficoll density gradient centrifugation using lymphocyte separation medium (Eurobio, Les Ulis, France). The CD34^+^ cells were then purified and selected using a magnetic-bead-based method (MiniMACS) with a CD34 Microbead kit (Miltenyi Biotec, Bergisch Gladbach, Germany) according to the manufacturer’s instruction. CD34^+^ cell purity was then determined by flow cytometry. Purified CD34^+^ cells were seeded into fibronectin-coated 12-well plates at either 5 × 10^5^ to 10^6^ cells/mL in a specific culture medium (US 2008/0153164 A1), allowing cardiovascular differentiation, as previously established in our laboratory, and supplemented with a combination of epigenetic-modifying agents [5,6,55]. Briefly, the cells were incubated in Iscove’s modified medium (IMDM), supplemented with 12.5% fetal bovine serum, 2.5% horse serum, 2 mM L-glutamine, (Eurobio, Les Ulis, France), 7 µM 1-thioglycerol (Sigma, St Quentin Fallavier Cedex, France), 1% penicillin/streptomycin, 1 ng/mL bone morphogenic protein 2 (BMP2), 5 ng/mL basic fibroblast growth factor (bFGF), 10 ng/mL vascular endothelial growth factor (VEGF), 20 ng/mL insulin-like growth factor-1 (IGF1), and 10 ng/mL bone morphogenic protein 4 (BMP4) (Miltenyi Biotec, Bergisch Gladbach, Germany). To induce cardiovascular differentiation, the cells were treated on days 1 and 2 with 2.5 mM VPA and 0.5 μM 5′AZA (STEMCELL Technologies, Grenoble, France), respectively. From day 3, the medium was renewed 3 times per week with previous medium containing 10 nM ascorbic acid and 1 ng/mL TGF-β1 (STEMCELL Technologies, Grenoble, France) until day 14. Adherent cells were then harvested, and their differentiation was evaluated by FCM, Real time RT-PCR, and microarrays.

### 4.3. Human Cardiomyocyte Progenitor Cell Culturing and Differentiation

Human cardiac biopsies were obtained from surgical waste from patients undergoing valve replacement surgery according to procedures approved by the Ethics Review Board of the local hospital. Human cardiomyocyte progenitor cells (CMPCs) were isolated, cultured in growth medium, and differentiated, as described by others [24]. Briefly, the tissue was minced and then digested in a solution containing collagenase for 2 h at 37 °C. After filtration and cardiomyocyte depletion, clonogenic isolation of the CMPCs was performed by limited dilution. Cells were expanded in M199:EGM-2 (3:1) (Lonza), supplemented with 10% FBS (Eurobio, Courtaboeuf, France) and 10 ng/mL bFGF. To induce cardiac differentiation, CMPCs were treated with 5 µM 5′AZA (Sigma Aldrich, St. Quentin Fallavier, France) for 72 h in IMDM: Ham’s F12 (1:1) supplemented with L-glutamine, penicillin/streptomycin, 2% horse serum, nonessential amino acids, and ITS (insulin, transferrin, and selenite) supplements (Invitrogen). This treatment was then followed by TGF-β1 stimulation at 1ng/mL and vitamin C (Sigma, St Quentin Fallavier Cedex, France) for 4 weeks (day 29), during which the differentiation medium was refreshed twice weekly.

### 4.4. FCM Analysis

The expressions of antigens associated with hematopoietic, cardiomyocyte, and endothelial lineages were evaluated. Human control and differentiated CD34^+^ cells were dissociated with 0.25% Trypsin-EDTA (Life Technologies, Courtaboeuf, France) for 4 min at 37 °C, washed, counted, suspended in PBS, and stained for 15 min at room temperature with different combinations of the following mouse anti-human monoclonal antibody surface markers: FITC-CD34 (130-113-178), APC-CD309 (KDR) (130-093-601), CD73-PE (130-112-060), CD133-PE (130-098-826), CD90-PE (130-117-537) (Miltenyi Biotec, Bergisch Gladbach, Germany), CD106-APC (305809) (Biolegend, Amsterdam, The Netherlands), CD105-VioBlue (130-112-320), CD117-PE-Vio770 (130-111-672), CD172a-PE-Vio770 (130-099-793), CD146-PE-Vio770 (130-099-957), CD344-PE-Vio770 (130-106-572), and CD31-APC (130-110-808) (Miltenyi Biotec). The cells were washed and suspended in PBS, and 7AAD was used to discriminate between living and dead cells. The different samples were then analyzed using a FacsCanto II instrument (Becton Dickinson Biosciences, Le Pont de Claix, France).

### 4.5. Immunofluorescence Labelling

The cells (CD34^+^ SCs) were cultured for differentiation in chamber slides, and then they were fixed in methanol for 10 min at 4 °C and blocked in 5% BSA for 1 h at room temperature before antibodies were added. Primary antibodies were specific for von Willebrand factor (vWF, Agilent Technologies, Les Ulis, France, A0082), Frizzled-4 (FZD4, R&D Systems, Noyal-Châtillon-sur-Seiche, France, MAB194), CD31 (Ozyme, Saint Quentin Yvelines, France, BLE349106), and Sarcomeric α-Actin (SIGMA, St. Quentin Fallavier, France, A7811). Secondary antibodies used were goat anti-Rabbit Alexa Fluor 488 (green) (Fischer Scientific, Saint Quentin Yvelines, France, A-11008), goat anti-rat Alexa Fluor 488 (green) (Fischer Scientific, Saint Quentin Yvelines, France, A-11006), and goat anti-mouse Alexa Fluor 555 (red) (Fisher Scientific, Saint Quentin Yvelines, France, A-21422 and A-21426). DAPI (CliniSciences, Nanterre, France) was used to visualize nuclei. Staining was analyzed by fluorescence microscopy.

### 4.6. RNA Isolation and Quantitative RT-PCR

Gene expression characterization of the differentiated cells was carried out in comparison with untreated human CD34^+^ and control cells. Total RNA was isolated and prepared using a RNeasy Plus Mini Kit (Qiagen, Courtaboeuf, France) and quantified (DeNovix, Wilmington, DE, USA). RNA was then reverse-transcribed (RT) into complementary DNA (cDNA) using IScript Supermix (Bio-Rad, Marnes-la-Coquette, France) according to the manufacturer’s instructions. Relative gene expression of the cDNA was assessed by real-time PCR using SsoAdvanced Universal SYBR Green Supermix (Bio-Rad) with a CFX96 Real-Time PCR System (Bio-Rad) and the indicated primers (Table 1).

All gene expression levels were normalized to the levels of HPRT1, PPIA, and RPLP0. Commercially available human cardiomyocytes and cardiomyocyte progenitor RNA (CliniSciences, Nanterre, France) were used as positive controls.

### 4.7. Microarray Experiments and Analysis

Microarray hybridization and data acquisition were performed by the HELIXIO genomic platform (Saint-Beauzire, France). Briefly, total RNA was extracted from UCB-CD34^+^ and UCB-CD34^+^ treated cells and, at different stages of differentiation, cardiac progenitors (CMPCs) and cardiomyocytes (4 pools of quadruplets) were also extracted. Then, the RNA sample quality was evaluated, and RNA integrity was controlled with a Bioanalyzer 2200 (Agilent Technologies, Les Ulis, France). Gene expression profile analysis was performed using a Low Input Quick Amp WT Labeling Kit with one color (Agilent Technologies, Les Ulis, France), and cDNA was reverse-transcribed from RNA samples and fluorescently labeled with cyanine 3. Each sample was hybridized with an Agilent SurePrint G3 Human Gene Expression 8x60K v2 Microarray and processed according to the manufacturer’s instruction. DNA microarrays were scanned using an Agilent G2505C DNA microarray scanner. The image files were extracted using Agilent Feature Extraction software. Data were then analyzed using Agilent GeneSpring GX software 12.0 (Agilent Technologies). The raw signals were log transformed and normalized using the percentile shift normalization method and the value was set at the 75th percentile. For each probe, the median of the log summarized values from all the samples was calculated and subtracted from each of the samples to become transformed to baseline. After normalization, data were filtered by the Agilent Feature Extraction software 11.5.1.1. Principal component analysis and unsupervised hierarchical clustering by Pearson’s distance measure on average linkage were performed with GeneSpring GX software 12.0. Gene significance was performed to determine differentially expressed genes through ANOVA (more than two groups) or unpaired *t*-test using the following parameters: *p* < 0.001, Benjamini–Hochberg false discovery rate as multiple testing correction, and Student-Newman-Keuls (SNK) as a post-hoc test for ANOVA. Analysis of differentially expressed genes: Bioinformatic analysis for enriched terms was performed using WEB-based Gene Set Analysis Toolkit (WebGestalt; http://www.webgestalt.org/, accessed on December 2015), which performs a hypergeometric test for the enrichment of gene ontology (GO) terms and Kyoto Encyclopedia of Genes and Genomes (KEGG pathways) in the selected genes, followed by the Benjamini-Hochberg (BH) method for multiple test adjustment (pAdj). 

The raw data were uploaded to GEO accession #: GSE221650.

### 4.8. Statistical Analysis

For microarray data, one-way ANOVA and *t*-tests were used to detect differential genes (significance was assigned for *p* < 0.001). *p* values were adjusted for false discovery rate. Please refer to microarray experimental and analysis for full details of Materials and Methods. For all other data, statistical analyses were performed using Student’s *t*-tests. Data are expressed as the means ± standard error of the mean (SEM). Statistical significance is indicated by asterisks; * *p* < 0.05, ** *p* < 0.01, *** *p* < 0.005, and **** *p* < 0.001. All the experiments were reproduced at least four times. 

## 5. Conclusions

To date, neither the molecular mechanism responsible for in vivo CD34^+^ SCs reprogramming to the cardiovascular pathway, nor clear, straight evidence of their direct implications in cardiac function recovery are known in detail. This knowledge is required and relevant, given the promise these cells show when intended for use as a cellular source to restore cardiac function in specific clinical therapy, which our study seems to support. 

Our findings provide evidence regarding possible CD34^+^ SCs initiation and reprograming to a cardiovascular destiny and help in the identification of the involved pathways and cardiac mesoderm markers expressed during early cardiovascular differentiation. Finally, these induced genes may be used as a blueprint to predict the cardiovascular potential of CD34^+^ SCs, making it possible to determine their ability to initiate their differentiation and benefits to their therapeutic application in cardiac regeneration.

## Figures and Tables

**Figure 1 ijms-24-09551-f001:**
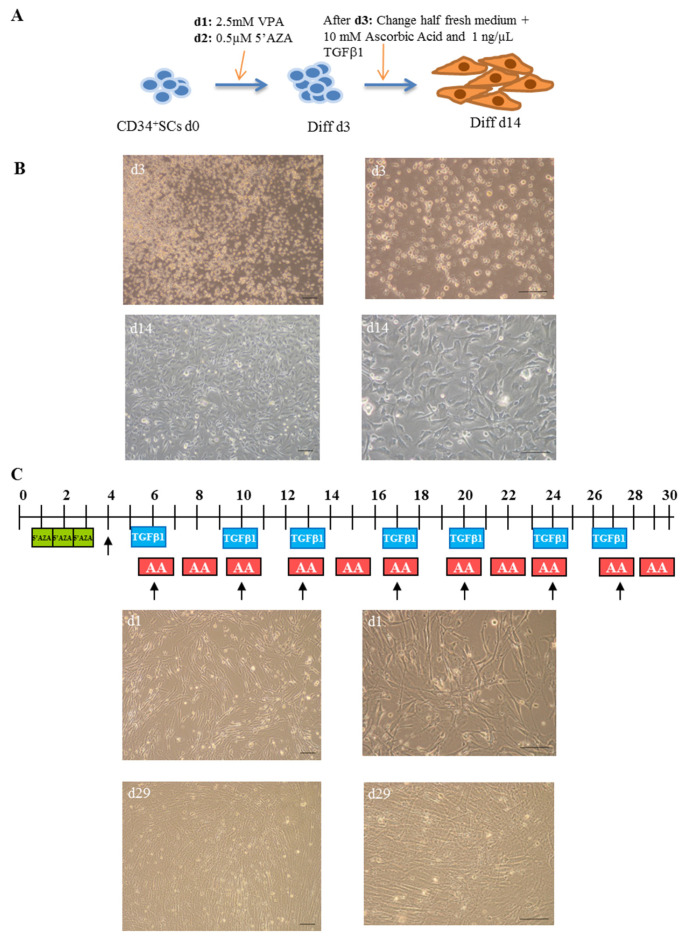
Cardiovascular differentiation of human CD34^+^ SCs and CMPCs: procedures and morphologies. (**A**) Schematic description of protocol used for cardiovascular differentiation of human CD34^+^ SCs to study gene expression profiles at day 0 (d0), d3, and d14 of differentiation. (**B**) Morphology of CD34^+^ SCs purified from human cord blood (H-UCB), as observed by optical microscopy on d3 and d14. (**C**) Cardiac differentiation procedure of cardiomyocyte progenitor cells (CMPCs) (upper panel) and their morphologies at d1 and d29 (lower panel). Bar scale: 100 µm.

**Figure 2 ijms-24-09551-f002:**
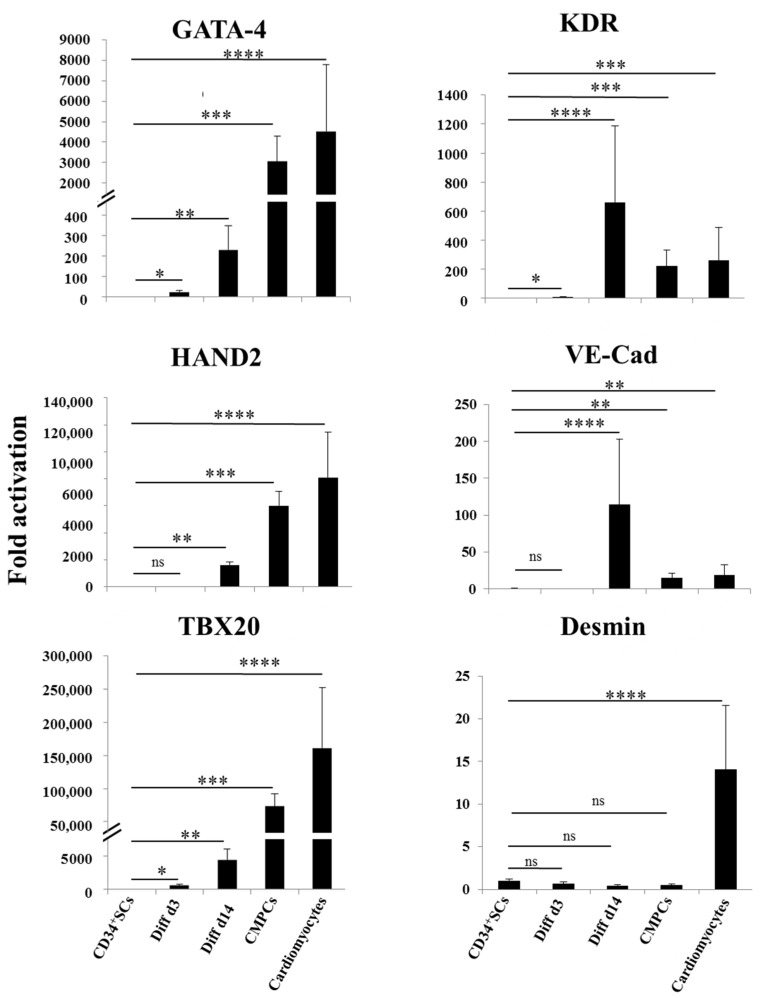
Cardiovascular gene expression. Relative expression analysis of indicated genes as determined by real time RT-PCR in CD34^+^ SCs, controls, and cells differentiated at d3 and d14, as well as CMPCs and cardiomyocytes. Data are presented as means ± SD of four individual experiments; Unpaired *t*-test, * *p* < 0.05, ** *p* < 0.01, *** *p* < 0.005 and **** *p* < 0.001 when compared to CD34^+^ SCs; ns: not significant.

**Figure 3 ijms-24-09551-f003:**
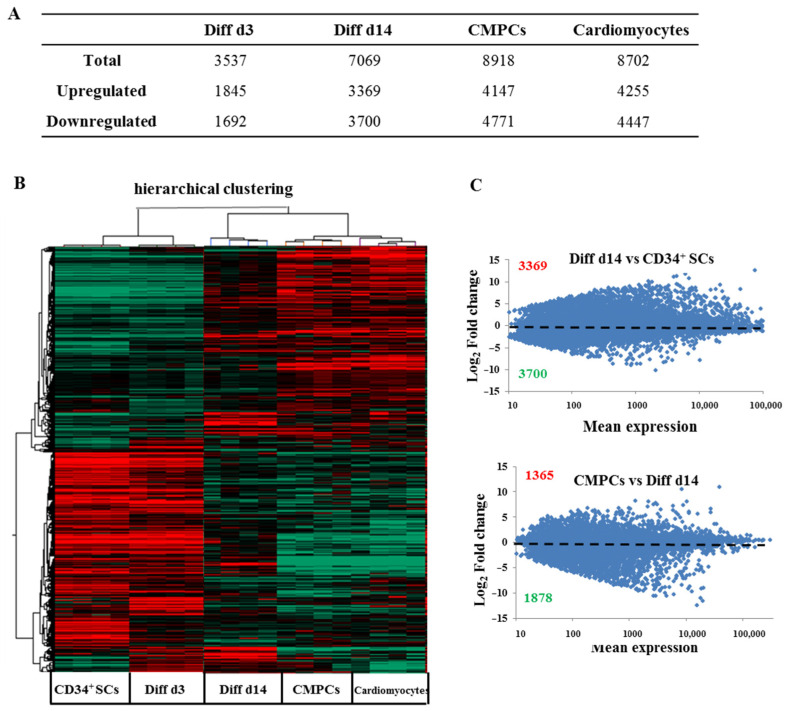
Global gene expression profiling of human CD34^+^ cells at different stages of differentiation. (**A**) Numbers of genes with expression changes significantly (*p* < 0.001) affected in the indicated samples versus CD34^+^ SCs. (**B**) Hierarchical clustering of differential expression profiles among human CD34^+^ cells and those at different stages of differentiation (Diff d3 and Diff d14), CMPCs and cardiomyocytes, based on Pearson correlation of significant (*p* < 0.001) differentially expressed genes among 20 samples. Upregulated genes and downregulated genes are, respectively, represented in red and green. (**C**) Differential expression profiles of gene expressions determined by microarray represented as scatter plot of the means of total gene expressions versus log_2_ fold-changes of Diff d14 cells compared to CD34^+^ SCs (upper panel) and of CMPCs compared to Diff d14 cells (lower panel) calculated from the means of four samples (Red represents the number of upregulated genes and green those downregulated). Statistical significance was calculated by ANOVA using the following parameters, *p* < 0.001, Benjamini-Hochberg’s correction and SNK test.

**Figure 4 ijms-24-09551-f004:**
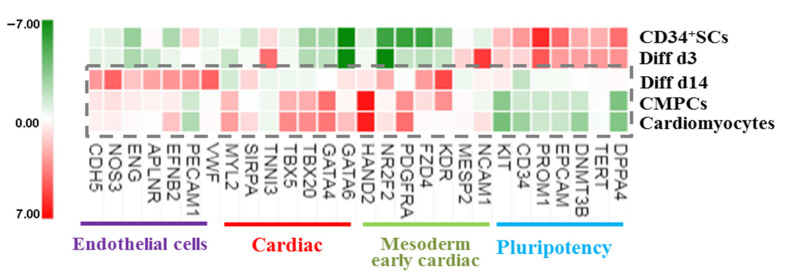
Heat map showing normalized expression variations in the indicated sets of genes in treated cells and differentiated CMPCs and cardiomyocytes relative to CD34^+^ SCs (green is decrease, and red is increase, relative to control). The color scale is shown at the bottom. Heat map shows average of normalized intensities for one condition (*n* = 4).

**Figure 5 ijms-24-09551-f005:**
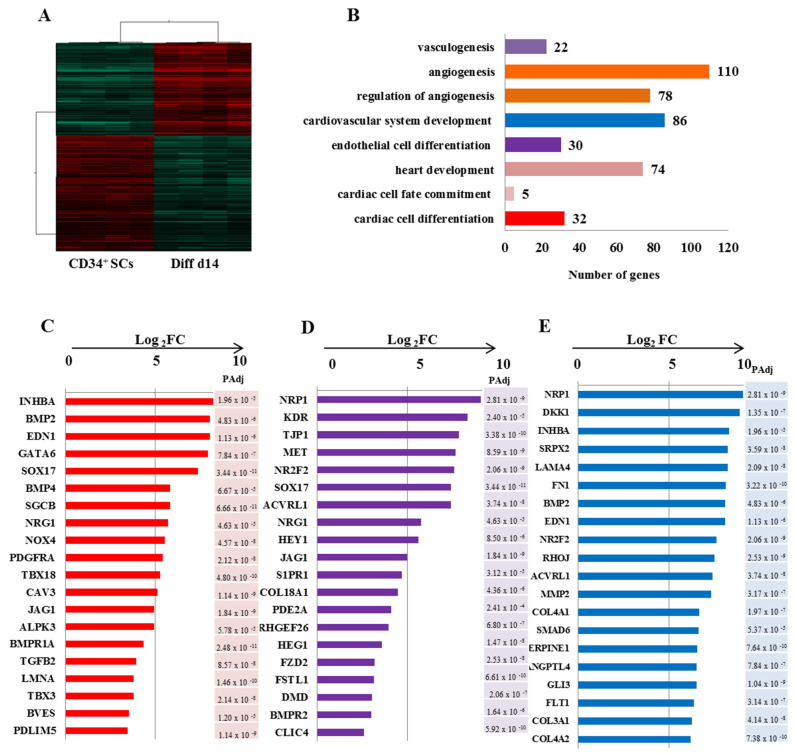
Differential gene expression profiling of Diff d14 vs. CD34 ^+^ SCs. (**A**) Hierarchical clustering carried out from 6837 probes by *t*-test (*p* < 0.001) on eight samples and based on Pearson correlation analysis. (**B**) Gene ontology analysis of the significantly up-regulated genes in Diff d14 cells versus CD34^+^ SCs and their classification according to their biological functions. The gene related to angiogenesis, cardiovascular system development, and heart development are overexpressed significantly in Diff d14 using *t*-test (significance was assigned for *p* < 0.001) with Benjamini–Hochberg’s correction. (**C**–**E**) Plots (Log_2_ FC) of top unregulated genes in Diff d14 vs. CD34^+^ SC for cardiac cell differentiation (**C**), endothelial differentiation (**D**), and cardiovascular cell development (**E**). A pAdj value < 0.001 is indicated for each gene and is calculated by adjusting the *p*-value with Benjamini Hochberg false discovery rate (FDR).

**Figure 6 ijms-24-09551-f006:**
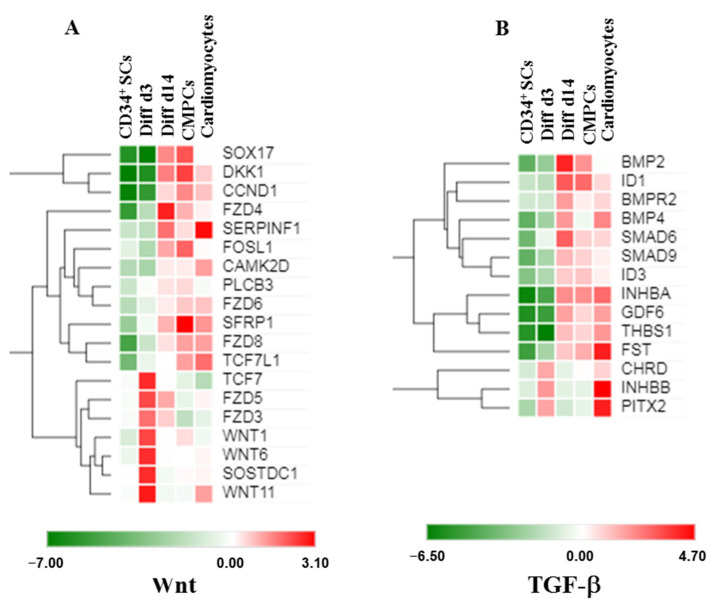
Heat maps showing expression levels of sets of genes involved in cardiovascular differentiation and Wnt (**A**) or TGF-β (**B**) signaling in CD34^+^ SCs, treated cells Diff d14, CMPCs, and cardiomyocytes (green is decrease and red is increase relative to control). The color scale is shown at the bottom. Heat maps show averages of normalized values for one condition (*n* = 4).

**Figure 7 ijms-24-09551-f007:**
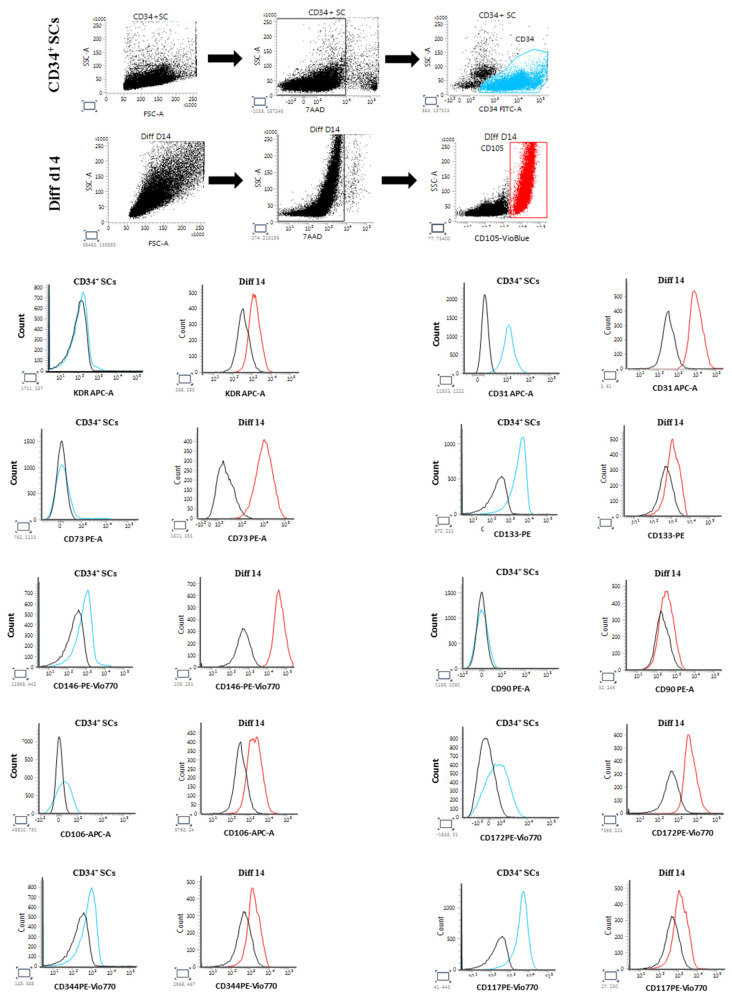
FCM analysis of cardiovascular receptor expressions in CD34-positive population of untreated CD34^+^ SCs (blue) and CD105-positive population of Diff d14 cells (red). The black histograms indicate the corresponding isotype signals.

**Figure 8 ijms-24-09551-f008:**
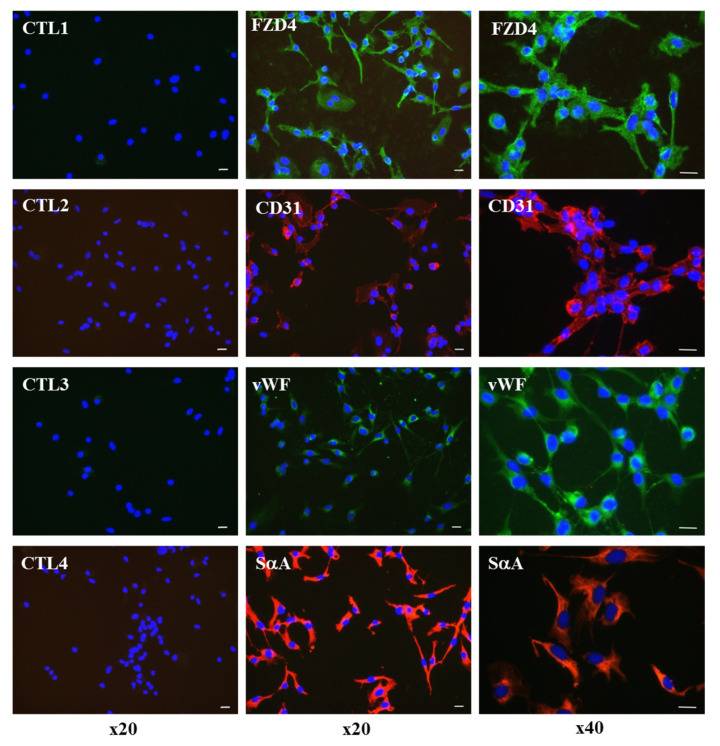
Immunofluorescence staining of endothelial and cardiac markers, FZD4, CD31, vWF, and sarcomeric α-actin (SαA) in Diff d14 cells. The left panels show the controls (CTL) with the secondary antibodies alone. Green is Alexa Fluor 488-stained, and red is Alexa Fluor 555-stained. DAPI was used to visualize nuclei (blue). The pictures display representative merged immunofluorescence images. Scale bar, 20 µm. (Original magnification, ×20 (left and middle panels), ×40 (right panel)).

**Figure 9 ijms-24-09551-f009:**
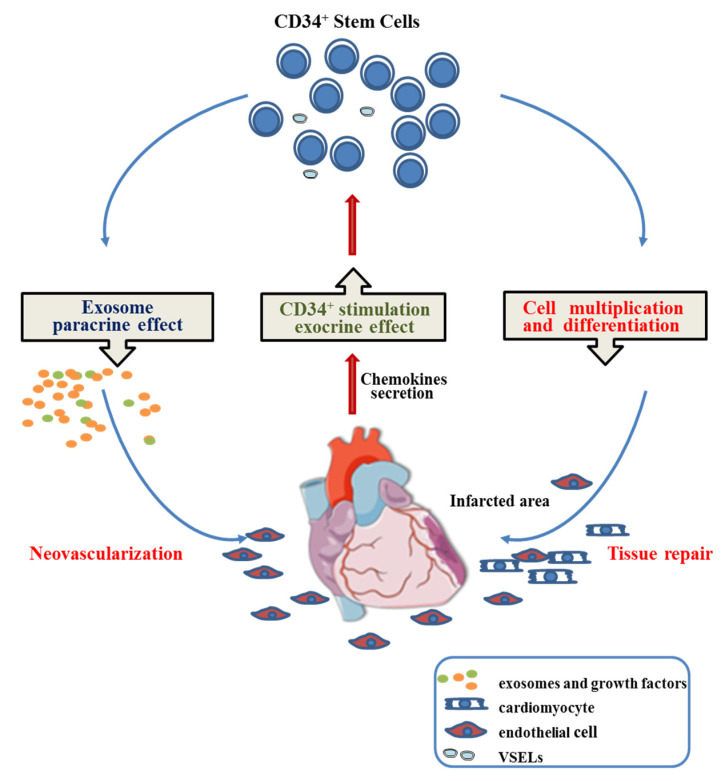
Proposed mechanism involved in CD34^+^ stem cell cardiac regenerative medicine.

**Table 1 ijms-24-09551-t001:** Forward and reverse oligonucleotides used for Real time RT-PCR.

Gene	Forward Sequence 5′-3′	Reverse Sequence 5′-3′
*GATA4*	AGATGCGTCCCATCAAGACGGA	ACTGACTGAGAACGTCTGGGACAC
*HAND2*	GCTACATCGCCTACCTCATG	CTGCTCACTGTGCTTTTCAAG
*TBX20*	CTGAGCCACTGATCCCCACCAC	CTCAGGATCCACCCCCGAAAAG
*KDR*	AGTGATCGGAAATGACACTGGA	GCACAAAGTGACACGTTGAGAT
*VE-Cad*	GTCCAACGGAACAGAAACATCCCT	GAGCATCATGAGCCTCTGCATCTT
*Desmin*	GATCAATCTCCCCATCCAGAC	GACCTCAGAACCCCTTTGC
*HPRT1*	TGCTTTCCTTGGTCAGGC	TCAAATCCAACAAAGTCTGGC
*PPIA*	CCGAGGAAAACCGTGTACTATTAG	TGCTGTCTTTGGGACCTTG
*RPLP0*	TATCACAGAGGAAACTCTGCATTC	GCCTTGACCTTTTCAGCAAGT

## Data Availability

The microarray data discussed in this publication have been deposited in NCBI’s Gene Expression Omnibus and are accessible through GEO Series accession GSE221650. The original contributions presented in the study are included in the article; further inquiries can be directed to the corresponding author.

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
