# Peer review of "Deciphering the Cardiovascular Potential of Human CD34+ Stem Cells"

_ijms, 2023, doi:10.3390/ijms24119551_

Round 1
Reviewer 1 Report
In their manuscript Aries et al. describe gene expression changes of human ex vivo CD34+ stem cells in the course of a differentiation protocol designed to direct the cells toward a cardiovascular lineage. Stating that there is no convincing evidence yet that CD34+ SC have the capability to differentiate into cardiac progenitors and cardiomyocytes, the authors compare the transcriptomes of these cell types with CD34+ cells after 3 and 14 days of in vitro differentiation and imply that cardiovascular priming and initiation to a myocardial lineage is possible.
To explore the potential of stem cells for cardiovascular regeneration is of high relevance for the field and especially studies on primary human cells are valuable. At first glance, the manuscript convinces by its very straightforward and clear structure with appropriate figures to illustrate the protocol and support the story. The results section reads really nice and although the study is solely observational the discussion section provides some proposed mechanism to embed the basic findings in a clinically relevant context. However, upon closer inspection the data presented here are not completely convincing either.
Some debatable aspects and concerns are as follows:
1) Generally, the sample sizes are low (which is understandable as the investigated cells are difficult to obtain) and the standard deviation is quite high as can be expected from such ex vivo cells. Yet, I assume that the authors only forgot to indicate the significant results in Figure 2 (the caption at least points at P values < 0.05), as differences between the groups are huge for all tested genes. Besides missing significances as well as the absence of any standard deviations for HAND2, what concerns me here is that the extremely high Fold changes might be rather artificial due to very low expression values for the initial CD34+SCs. Apart from stating that expression data were normalized to levels of three housekeeping genes, the authors do not give information on the analysis procedure to get the “Fold activation”, which I just assumed to be similar to Fold change. Since the figure suggests that expression levels of the cardiac genes for CD34+SCs are at least near the detection limit, I would be interested in the original Ct-values and the method used for the calculations, since using calculated transcript numbers around zero will result in inflated Fold changes for those samples that actually yielded any signal. Without the raw data given as supplements it is hard for me to fully trust these results.
2) Even if the degree of gene expression changes is inflated by the calculation method, the differences between Diff d14 and CMPCs (as well as cardiomyocytes) seem to be significant, which puts the actual similarity to cells from the cardiovascular lineage into perspective. In addition to the two positive controls, some sort of negative control, for example CD34+ cells differentiated towards the lymphoid or myeloid lineage, would have been required to assess whether upregulation of the presented genes is indeed an evidence for cardiac lineage commitment or to some extent “random noise” during the transition from multi potent stem cells to pluripotent cells of some lineage.
3) Experimental Methods are appropriate and clearly described, however, when it comes to the statistical analysis more details should be provided. To be transparent it should be stated exactly when Student´s t-test was used, as it actually is neither the recommended method to analyze differences between multiple groups (here ANOVA would be an appropriate option) nor for the analysis of transcriptome data, which demand an appropriate multiple comparison correction such as the Benjamini Hochberg procedure. 50% of analyzed genes were found to be significantly regulated, which appears quite much to me, but from comparison of Figure 2 and Figure 4, data are in accordance with the PCR results providing some validity. Yet, again expression levels of cardiac genes are minor compared to CMPCs and Cardiomyocytes, while genes typical of endothelial cells are severely upregulated (which again is in accordance with the PCR data). Cardiac-specific transcription factors are reported to be “enriched at a relatively significant level” in Diff d14 as well as some genes declared as early cardiac markers. However, FZD4 and NR2F2 are mainly expressed by endothelial cells (see https://www.proteinatlas.org/), thus these data are not sufficient to demonstrate cardiac differentiation. The data do convincingly demonstrate a differentiation into endothelial cells, hence confirming numerous other studies indicating CD34+ cells as endothelial progenitor cells (as cited by the authors themselves in the introduction), but apparently the novelty of this finding is rather limited.
4) Aware of their contribution in cardiac regeneration, the study is supposed to aim at “deciphering the cardiovascular potential” of CD34+ SC, but in my opinion further aspects would need to be considered to ensure comprehensive and in-depth insights. For example cells were treated with the DNA methylation inhibitor 5-azacytidine. In general, but even more so in this case, further experiments to follow the epigenetic changes due to the treatment and during the differentiation process would be strongly recommended. Methylation of relevant promotors as well as differentially methylated regions in general would provide a layer of information that is completely missing here.
5) The authors identified the Wnt and the TGF-ß pathway to be regulated during differentiation. As these are well-known pathways in cardiomyocyte differentiation, this seems to be in line with the idea that CD34+ SC can be directed toward a cardiovascular lineage. However, on the one hand the authors treated the cells with TGF-ß, which renders changes in the respective signal pathway to be expected and on the other hand a very recent single cell study demonstrated that these pathways are implicated in the differentiation of CD34+ SC to fibroblasts (Du et al. Single cell and lineage tracing studies reveal the impact of CD34+ cells on myocardial fibrosis during heart failure. Stem Cell Res Ther. 2023 Feb 20;14(1):33. doi: 10.1186/s13287-023-03256-0.). Accordingly, the regulation of these pathways cannot be considered as a solid evidence for the commitment of the CD34+ SC toward a cardiovascular lineage.
In summary, the study findings are relevant for the research field of cardiac regeneration and the microarray data the authors provide would be very valuable especially due to the rare cell source. However, although all parts of the manuscript are written properly and some results imply a cardiac priming, the power of persuasion of the presented data is still limited. It is very apparent that the authors try not to overstate their data, which is appreciated by this reviewer. However, to accomplish the objective of the paper to decipher the cardiovascular capacity, the study design would need to be more comprehensive/in depth (characterizing epigenetic changes, transcription and protein levels, plus potentially some functional readouts), as the fact that CD34+ cells have the capability to differentiate into (cardiac) endothelial cells is not novel at all.
Author Response
"Please see the attachment."

Reviewer 2 Report
The manuscript (MS) entitled “Deciphering the cardiovascular potential of human CD34+ stem cells” by Anne Aries and colleagues, presents results of study in human CD34+ stem cells (SCs) and their differentiation into the cardiogenic/vascular cells. The study was carried out in cell cultures of human CD34+ SCs from human umbilical cord blood and in human cardiomyocyte progenitor cells (CMPCs) obtained from tissue samples from cardiac biopsies from patients undergoing valve replacement surgery. The reported findings indicate that CD34+ SCs may differentiate into early cardiac and vascular lineage of cells, possibly involving WNT and TGF-beta signalling pathways.
The MS is concise and well written. The bioethical issues related to harvesting human cells and carrying out experiments are taken care of, and the study received ethical approvals.
I find the MS interesting and going back to bench after not so successful clinical applications of stem cells/progenitor cells in cardiovascular medicine.
The topic is clinically relevant, as the use of progenitor/stem cells for treatment of failing heart in heart failure and/or myocardial infarction is still under debate and investigations in this field have been ongoing. Use of cells sourced from humans increases translational meaning of the reported results. In my view the paper provides incremental, however important, information to the field that is worth disseminating to the research community.
The main flaw of the MS is insufficient description of results in terms of statistical analysis. Please, find my additional comments below.
Major points:
1. Were there any tests carried out for primary antibody specificity? Was unspecific binding of the secondary antibodies to the epitopes evaluated with negative control (i.e. staining without the use of primary antibodies)?
2. Statistical analysis – the authors used parametric tests (student t-test, ANOVA). Please, confirm that all data fulfilled the criteria of normal distribution and normality or re-evaluate the data that does not fulfil the above criteria with proper statistical tools (for example Krukal-Wallis test instead of ANOVA, or Mann–Whitney U test instead of t-test; but other statistical approaches are also welcome). Furthermore, some statistical tests/techniques used in the study are not mentioned/explained in the Statistics sub-section in the Methods (for example hierarchical clustering by Pearson correlation used for analysis of genes’ transcriptional signatures)
3. The description of the results does not refer in any way to the statistical analysis. It is not clear for a reader which findings were statistically analyzed and if the differences are meaningful or not. The results should include some statistical comment in addition to descriptive term. For example: “The expression of desmin remained moderate in Diff 3 and 14” – Line 125 – does not indicate if expression was significantly changed. Does moderate mean significant, or insignificant, or is it only “visual inspection” of results without statistical analysis? This comment is pertinent to all results – please, clarify if statistical analysis was carried out, what statistical test were used and if they were significant for all results.
4. Figure 2 – please, clarify if there are no significant differences between cell types/treatments. The figure caption provides “*P < 0.05” (line 132), however, no significance is indicated on the graphs.
5. Other Figures – were there any statistics done for results presented? If they were insignificant, this information should be also provided.
6. Please, provide catalogue numbers of anti-bodies used in the study.
7. I suggest including a paragraph on the limitations of the study, the main one being that WNT and TGF-beta pathways were not blocked to confirm their role in the differentiation of the CD34+ SCs.
Minor points:
1. Several abbreviations used in the MS are explained in the Methods, however, this section is at the very end of the paper. The abbreviations appear earlier in the Results and in the Discussion – they be introduced and explained in the Results for clarity. Some examples: VPA, AZA, PMPCs, FCM etc. – more to be found in the MS.
2. Figure 4 and placement of its description are misaligned.
3. Figure 8 – please, confirm that panels on the left and on the right show CD34+ cells on day 14 of differentiation. I suggest clarifying this also in the figure caption.
Author Response
"Please see the attachment."

Round 2
Reviewer 1 Report
The authors addressed the most major points concerning statistics and figures, which definitely improved the manuscript. In particular, severely more details about the statistical approaches for the Microarray experiments were added. However, I personally still feel that a multiple comparison by Anova would be the more appropriate statistical method for the analysis of marker gene expression by PCR, as also the differences between in vitro differentiated SCs and CMPCs are relevant.
More general recommendations such as implementing negative controls and extending the study with some epigenetic approaches such as ChIP-seq were appreciated by the authors but not considered for this manuscript. I agree with the authors that their data are already worth sharing with the scientific community, it´s on the editor to decide whether the relevance of this manuscript is high enough for this journal. The study is sound and the manuscript is well written, accordingly, I would have no objections to publication, looking forward to the extension of the authors work into the field of epigenetics.
Reviewer 2 Report
The authors addressed my all major points related to statistics, figures and methods. The MS is significantly improved. In my view it is publishable now.
Minor points:
There are minor typographical errors that require correction:
Line 279: "Heat maps shows average of normalized values for one condition (n=4)." should be:
"Heat maps show average of normalized values for one condition (n=4)."
Line 314-315: "second antibodies" should be "secondary antibodies"